# Intelligent Switching in Reset-Free RL

**Darshan Patil** *
Mila, Université de Montréal

**Janarthanan Rajendran** †
Dalhousie University

**Glen Berseth**
Mila, Université de Montréal
Canada CIFAR AI Chair

**Sarath Chandar**
Mila, École Polytechnique de Montréal
Canada CIFAR AI Chair

## Abstract

In the real world, the strong episode resetting mechanisms that are needed to train agents in simulation are unavailable. The *resetting* assumption limits the potential of reinforcement learning in the real world, as providing resets to an agent usually requires the creation of additional handcrafted mechanisms or human interventions. Recent work aims to train agents (*forward*) with learned resets by constructing a second (*backward*) agent that returns the forward agent to the initial state. We find that the termination and timing of the transitions between these two agents are crucial for algorithm success. With this in mind, we create a new algorithm, Reset Free RL with Intelligently Switching Controller (RISC) which intelligently switches between the two agents based on the agent's confidence in achieving its current goal. Our new method achieves state-of-the-art performance on several challenging environments for reset-free RL. *

## 1 Introduction

Despite one of reinforcement learning's original purposes being as a way to emulate animal and human learning from interactions with the real world (Sutton & Barto, 2018), most of its recent successes have been limited to simulation (Mnih et al., 2015; Silver et al., 2016; Team et al., 2023). One of the reasons for this is that most work trains agents in episodic environments where the agent is frequently and automatically reset to a starting state for the task. Environment resetting, while simple to do in a simulator, becomes much more expensive, time consuming, and difficult to scale for real world applications such as robotics, as it requires manual human intervention or the use of highly specialized scripts.

Our current algorithms are designed around the ability to reset the environment, and do not transfer to settings without resets (Co-Reyes et al., 2020). Resets allow the agent to practice a task from the same initial states, without needing to learn how to get to those initial states. This revisiting is critical for RL agents that learn through trial and error, trying different actions from the same states to learn which are better. Resets also allow the agent to automatically exit problematic regions of the state space. It is often much easier to enter certain regions of the state space than it is to exit them (e.g., falling down is easier than getting back up).

Recent works have started to explore learning in environments where automatic resets are not available in a setting known as reset-free or autonomous RL. A common approach for such settings is to have the agent switch between a forward controller that tries to learn the task, and a reset controller that learns to reset the agent to favorable states that the forward controller can learn from (Eysenbach et al., 2017; Han et al., 2015; Zhu et al., 2020; Sharma et al., 2021a; 2022). Crucially, the part of these algorithms that switches between the controllers has gone understudied.

This paper explores how we can improve performance by switching between controllers more intelligently. Prior works have not established a consistent strategy on *how* to bootstrap the value

---

*Correspondence to darshan.patil@mila.quebec
†Work done during Postdoc at Mila, Université de Montréal
*Code available at https://github.com/chandar-lab/RISC

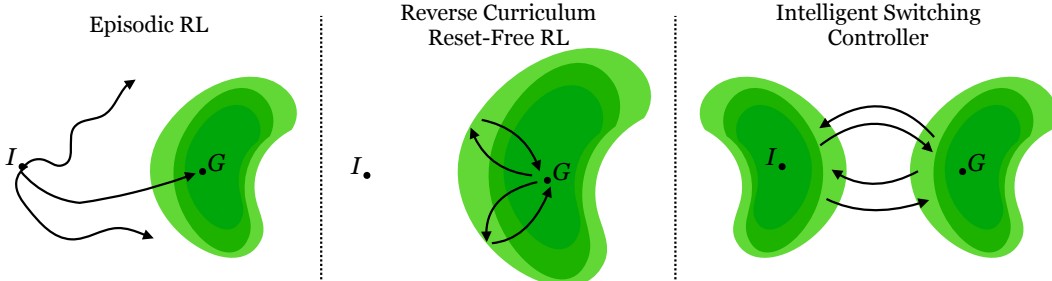

Figure 1: In RL, the agent usually starts learning about the task in areas around rewarding states, and eventually propagates the learning to other parts of the state space. (Left) In episodic learning, the agent starts its trajectories at a state in the initial state distribution. Through exploration, it might find a trajectory that produces a reward, but might struggle to reach that goal state again, particularly on sparse reward or long horizon tasks. (Center) A common approach in Reset-Free RL is to build a curriculum outward from the task's goal states. While this allows the agent to frequently visit rewarding states, it also means the majority of the agents experience will be generated in areas it has already learned. (Right) RISC switches directions when it feels confident in its ability to achieve its current goal (both in the forward (task's goal states) or the backward direction (task's intial states)). This not only reduces the time spent in already explored regions of the state space, but also reduces the average distance to the goal which makes it easier for the agent to find high value states.

of last state before the agent switches controllers. Bootstrapping is when the agent updates some value estimate using an existing value estimate of a successor state (Sutton & Barto, 2018). It is a common idea used across many RL algorithms. Pardo et al. (2022) empirically established that different bootstrapping strategies for the last state in a trajectory can result in different policies. We extend this result by analyzing why different bootstrapping strategies can result in different optimal policies. We show that bootstrapping the last state in the trajectory is crucial for reset-free RL agents to maintain consistent learning targets, and doing so greatly improves their performance.

Another underexplored aspect of the switching mechanism is learning *when* to switch. Because reset-free RL is a unique setting in that there is no episode time limit imposed by the environment while training, the duration of the agent's controllers' trajectories becomes a parameter of the solution method. Prior work generally uses a fixed time limit, but can learning when to switch also help the agent learn? For example, if the agent enters a part of the state space where it already knows how to accomplish its current goal, gathering more experience in that area is unlikely to help it further in learning the task. Switching controllers and thus the goal of the agent can allow the agent to learn more efficiently by gathering experience for states and goals it has not mastered yet.

Based on these two ideas, we propose Reset Free RL with Intelligently Switching Controller (RISC), a reset-free RL algorithm that intelligently switches between controllers. To reduce the amount of time spent in parts of the state space that the agent has already learned well in, we learn a score corresponding to the agent's ability to reach its current goal (the current goal could be either in the forward or the backward direction). The agent then switches directions with probability proportional to that score. This allows the agent to maximize the experience generation in areas of the state space that it still has to learn more from (Figure 1 right). We evaluate our algorithm's performance on the recently proposed EARL benchmark (Sharma et al., 2021b). The benchmark consists of several robot manipulation and navigation tasks that need to be learned with minimal environment resets, and we show that our algorithm achieves state-of-the-art performance on several of these reset-free environments.

## 2 RELATED WORK

RL algorithms developed for episodic settings often fail on even simple tasks when trained in reset-free environments (Co-Reyes et al., 2020; Sharma et al., 2021b). Several settings have been proposed to address the shortcomings of episodic RL environments including continuing RL (Khetarpal et al., 2022), which tries to maximize the training reward over the lifetime of the agent, and single lifetime

RL (Chen et al., 2022) which tries to accomplish a task just once in a single lifetime. Other works have explored the problem of learning to practice in separate training environments in episodic settings (Rajendran et al., 2019). Our setting is different in that the agent practices in a non-episodic environment, but is evaluated on a task in an episodic environment. Several approaches have been proposed to enable learning without resets, including unsupervised skill learning (Xu et al., 2020; Lu et al., 2021) and framing the problem as a multi-task learning problem (Gupta et al., 2021; 2022).

A simple approach to learn in this setting is Forward Backward RL (FBRL), which alternates between a forward controller that tries to accomplish the task and a reset controller that tries to recover the initial state distribution (Eysenbach et al., 2017; Han et al., 2015). Other works explore different reset strategies, such as R3L where the reset controller tries to reach novel states (Zhu et al., 2020) or MEDAL which tries recover the distribution of states in demonstration data (Sharma et al., 2022; 2023). VapRL uses demonstration data to make a reset controller that builds a reverse curriculum for the forward controller (Sharma et al., 2021a). IBC, a concurrent work, uses optimal transport to create a curriculum for both the forward and reset agents without demonstrations (Kim et al., 2023).

Early resets have been used in the context of safe reset-free RL where the reset policy is triggered when the forward policy is about to do something unsafe (Eysenbach et al., 2017; Kim et al., 2022). Notably, these resets happen from the opposite direction compared to our method (see Center vs Right in Figure 1). Instead of taking the agent back to areas it is confident in, as these methods do, our method aims to maximize the amount of experience gathered in areas of the state-goal space where the agent has not learned.

## 3 PRELIMINARIES

In this section, we define the problem of reset-free RL. Consider a goal-conditioned Markov Decision Process (MDP): $\mathcal{M} \equiv (\mathcal{S}, \mathcal{A}, \mathcal{G}, \mathcal{T}, r, \rho, p_g, \gamma)$ where $\mathcal{S}$ denotes the state space, $\mathcal{A}$ denotes the action space, $\mathcal{G}$ denotes the goal space, $\mathcal{T}(\cdot|a, s)$ is a transition dynamics function, $r(s, a, g)$ is a reward function based on goal $g$, $\rho$ is the initial state distribution, $p_g$ is the desired goal distribution, and $\gamma \in [0, 1]$ is a discount factor denoting preference for long term rewards over short term rewards. In episodic RL, the objective is to find a policy $\pi$ that maximizes

$$J(\pi) = \mathbb{E}_{s_0 \sim \rho, g \sim p_g, a_t \sim \pi(s_t, g), s_{t+1} \sim \mathcal{T}(\cdot|a, s)} \left[ \sum_t \gamma^t r(s_t, a_t, g) \right]. \tag{1}$$

In this formulation, sampling $s_0 \sim \rho$ starts a new episode and "resets" the environment. Episodic RL assumes regular access to resets, usually happening every few hundred or thousand timesteps.

Our problem setting of reset-free RL, formalized as the "deployment" setting in Sharma et al. (2021b), involves maximizing the same objective as above. The main difference between the episodic setting and the reset-free setting is that the reset-free setting allows for much less frequent resetting of the environment during training (anywhere from once every few hundred thousand steps to just once at the beginning of training). In the episodic case, because the agent is reset to the initial state distribution frequently, it can directly evaluate the objective and much more easily get a signal for how it is performing. In the reset-free setting, because the agent is reset to the initial state distribution infrequently, the agent has to improve its performance on the objective while getting much less direct feedback on its performance.

## 4 RESET FREE RL WITH INTELLIGENTLY SWITCHING CONTROLLER

In this section, we describe Reset Free RL with Intelligently Switching Controller (RISC) and its components. Section 4.1 discusses the importance of proper bootstrapping when switching controllers in Reset-Free RL, Section 4.2 describes RISC's early switching mechanism, and Section 4.3 summarizes our entire approach.

### 4.1 ON BOOTSTRAPPING FOR RESET-FREE RL

In this section, we highlight the importance of proper bootstrapping when switching controllers in Reset-Free RL. For any policy $\pi$, the value function $V^\pi(s)$ measures the expected discounted sum of

rewards when taking actions according to $\pi$ from state $s$:

$$V^\pi(s) = \mathbb{E}_\pi \left[ \sum_{t=0}^\infty \gamma^t R(s_t, a_t) | s_0 = s, a_t \sim \pi(\cdot|s_t) \right]. \tag{2}$$

For brevity and clarity of exposition, we focus our analysis on the value function, but our analysis can be extended to state-action value functions with basic modifications. The value function satisfies the Bellman equation (Bellman, 1957):

$$V^\pi(s) = \mathbb{E}_\pi \left[ R(s, a) + \gamma V^\pi(s') | s, a \sim \pi(\cdot|s), s' \sim \mathcal{T}(\cdot|s, a) \right]. \tag{3}$$

Most RL algorithms involve learning either the value function or the state-action value function, usually through temporal difference (TD) learning (Sutton & Barto, 2018). The main mechanism of TD learning is bootstrapping, where the value of a state is updated using the value of a successor state with an update mirroring the Bellman equation. When learning in an episodic environment, it is common to have time limits that end the episode before the agent has reached a terminal state. These early resets are useful because they can diversify the agent's experience. The value of terminal states is generally not bootstrapped, as by definition, the value of a terminal state is zero and it has no successors. Pardo et al. (2022) empirically showed that bootstrapping the value of the last state in a trajectory that ended because of a timeout can result in not only a different policy than not bootstrapping, but also better performance and faster learning. We investigate why this is the case and discuss the implications for reset-free RL.

Let us denote the strategy where the value function is learned by bootstrapping the last state in a trajectory that ended in a timeout as *timeout-nonterminal* and the strategy where such states are not bootstrapped as *timeout-terminal*. Based on equation 3, the loss for the timeout-nonterminal strategy corresponds to

$$\mathcal{L}_n(\theta) = \mathbb{E}_{s \sim \xi, s' \sim P_\pi(\cdot|s)} \left[ \| V_\theta^\pi(s) - (r + \gamma V_\theta^\pi(s')) \| \right]. \tag{4}$$

Here $\xi$ corresponds to the weighting of different states according to the experience that is gathered by the agent, and $P_\pi(\cdot|s)$ is the transition distribution under policy $\pi$. The loss for the timeout-terminal strategy corresponds to

$$\mathcal{L}_t(\theta) = \mathbb{E}_{s \sim \xi, s' \sim P_\pi(\cdot|s), d \sim \kappa} \left[ \| V_\theta^\pi(s) - (r + \gamma(1-d) V_\theta^\pi(s')) \| \right] \tag{5}$$

The loss uses the same terms as the timeout-nonterminal strategy, except for a binary random variable $d$ drawn from some process $\kappa$ that represents whether the episode ended because of a timeout or not.

In episodic RL, $\xi$ is the stationary distribution over states for policy $\pi$, and depends only on $\pi$, the initial state distribution $\rho$, and the transition dynamics function $\mathcal{T}$. $\kappa$ is the rule that returns 1 when the trajectory gets over a certain length, and thus $\mathbb{E}[d|s]$ only depends on $\pi$, $\rho$, $\mathcal{T}$, and $\kappa$. The difference in the two loss terms results in the magnitude of the value of some states being reduced in the timeout-terminal strategy, depending on $\mathbb{E}[d|s]$. This explains why different policies can emerge from the two strategies.

In reset-free RL, these parameters become more complicated. We can think of there being two policies: the forward policy $\pi_f = \pi(\cdot|s, g_f)$ and the reset policy $\pi_r = \pi(\cdot|s, g_r)$ where $g_f$ and $g_r$ are forward goals and reset goals respectively. Examining the bootstrap loss for the forward policy, $\xi$ is now no longer dependent on just $\pi_f$, $\rho$, $\mathcal{T}$, and forward timeout process $\kappa_f$ but also on $\pi_r$ and reset timeout process $\kappa_r$. Furthermore, $\kappa_f$ and $\kappa_r$ are no longer static rules provided by the environment, but are decision rules based on the agent. Despite this, the timeout-nonterminal loss in Equation 4 at least maintains the same targets for the value of each state. $\mathbb{E}[d|s]$, however, is now dependent on $\pi_f$, $\pi_r$, $\rho$, $\mathcal{T}$, $\kappa_f$, and $\kappa_r$. This means that when using the timeout-terminal bootstrap loss (Equation 5), *the value function for the forward policy depends on the actions of the agent's reset policy and vice versa*. Thus, in order to maintain consistent targets while training, RISC uses the timeout-nonterminal loss when switching controllers.

## 4.2 Building a Curriculum with Intelligent Switching

Using a curriculum of tasks through the course of training has been shown to accelerate learning in reinforcement learning (Portelas et al., 2020; Florensa et al., 2017; Sukhbaatar et al., 2018). A common approach to building these curricula is to start with tasks with initial states near the goal, and expand outwards (Florensa et al., 2017), known as a reverse curriculum.

**Algorithm 1:** Reset Free RL with Intelligently Switching Controller (RISC)

---

**Input :** Trajectory switching probability: $\zeta$
$s, g = $ env.reset()
$t = 0$
$check\_switch = $ random() $< \zeta$
**while** *True* **do**
    $a = $ agent.act$(s, g)$
    $s', r = $ env.step$(a)$
    agent.update$(s, a, r, s', g)$
    $t = t + 1$
    **if** should_switch*(t,*
    agent.$Q_f$, s', g,
    *check_switch)* **then**
        $g = $ switch_goals()
        $t = 0$
        $check\_switch = $
        random() $< \zeta$
    **end**
    $s = s'$
**end**

**Algorithm 2:** Switching function

---

**Input:** Minimum trajectory length: $m$
        Maximum trajectory length: $M$
        Conservative factor: $\beta$
        Current state, goal: $s, g$
        Length of current trajectory: $t$
        Success Critic: $F$
        Whether to perform switching check:
        $check\_switch$
**if** $s == g$ **then**     // Switch if reached goal
    **return** *True*
**else if** $t \geq M$ **then**// Truncate after $M$ steps
    **return** *True*
**else if** $t < m$ **then**     // Min. Length Check
    **return** *False*
**else if** $check\_switch$ **then**
    $c = F(s, g)$ ;     // agent competency
    $\lambda = c \times (1 - \beta^t)$ ;     // P(switch)
    **return** $random() < \lambda$
**else**
    **return** *False*
**end**

Some Reset-Free RL methods explicitly build a reverse curriculum (Sharma et al., 2021a); other methods while not building a curriculum still follow the paradigm of always returning to the goal state. While reverse curriculums are a popular idea, we ask if they are the best approach for reset-free RL, as they can involve the agent spending a significant amount of environment interactions in parts of the state space it has already mastered (Figure 1, middle). Instead, we hypothesize that reset-free RL can benefit from a new type of curriculum that works from the outside in.

Our algorithm is based on the intuition that if the agent is already competent in an area of the state space, then it does not need to continue generating more experience in that part of the state space. Instead, it could use those interactions to learn in parts of the state space it still needs to learn, leading to more sample efficient learning. To that end, we introduce Reset Free RL with Intelligently Switching Controller (RISC). Similar to Forward-Backward RL (Han et al., 2015; Eysenbach et al., 2017), RISC alternates between going forward towards the task goal states and resetting to the task initial states. Unlike Forward-Backward RL and other reset-free methods, RISC switches not only when it hits some predefined time limit or when it reaches its current goal (in the forward or the backward direction), but also when it is confident that it can reach its current goal. This allows it to spend more time exploring regions of the state-goal space that it is unfamiliar with (Figure 1, right).

### 4.2.1 EVALUATING THE AGENT'S ABILITY TO ACHIEVE ITS GOALS

To build our intelligent switching algorithm, we need a way to estimate if an agent is competent in an area of the state space. A natural way to do this is to incorporate some notion of how many timesteps it takes the agent to achieve its goal from its current state. Thus, we define the competency of a policy $\pi$ in state $s$ on goal $g$ as

$$F_\pi(s, g) = \gamma_{sc}^t, t = \min\{t' : s_{t'} = g, s_0 = s, a_t \sim \pi(s_t, g), s_{t+1} \sim \mathcal{T}(\cdot|a, s)\}, \quad (6)$$

where $\gamma_{sc} \in [0, 1]$ is the success critic discount factor and $t$ represents the minimum number of steps it would take to get from state $s$ to goal $g$ when following policy $\pi$. This metric has several appealing qualities. First, it is inversely correlated with how many timesteps it takes the agent to reach its goal, incorporating the intuition that becoming more competent means achieving goals faster. It also naturally assigns a lower competency to states that are further away from a goal, which incorporates the intuition that the agent should be more uncertain the further away it is from its goal. Finally, this metric can be decomposed and computed recursively, similar to value functions in RL. This allows us to use the same techniques to learn this metric as we use to learn value functions.

We assume the ability to query whether a given state matches a goal, but *only* for goals corresponding to the task goal distribution or the initial state distribution. As the goal and initial state distributions

represent a small portion of states, learning such a function is not an unreasonable requirement even for real-world applications. We can then train a *success critic* ($Q_F$) to estimate $F_\pi(s, g)$. Specifically, given a success function $f : \mathcal{S} \times \mathcal{G} \to \{0, 1\}$ that determines if state $s$ matches goal $g$, we train a network to predict

$$Q_F(s_t, a_t, g) = \mathbb{E}_\pi \left[ f(s_{t+1}, g) + (1 - f(s_{t+1}, g)) \gamma_{sc} \hat{Q}_F(s_{t+1}, a_{t+1}, g) \right], \qquad (7)$$

where $Q_F$ and $\hat{Q}_F$ are the success critic and target success critic respectively. The structure and training of the success critic is similar to that of value critics used in Q-learning or actor critic algorithms. In fact, it is exactly a Q-function trained with rewards and terminations coming from the success function. The value of this success critic should always be between 0 and 1. At inference time, we compute the value of the success critic as $F_\pi(s, g) = \mathbb{E}_{a \sim \pi(s,g)}[Q_F(s, a, g)]$.

Because we want the success critic's value to mirror the agent's ability to actually achieve the goal, we train $Q_F$ on the same batch of experience used to train the agent's policy $\pi$ and critic $Q$ and set $\gamma_{sc}$ to the value of the agent's critic's discount factor.

### 4.2.2 MODULATING THE SWITCHING BEHAVIOR

Because $F(s, g) \in [0, 1]$, it might be tempting to use it directly a probability of switching. It's important to note that the output of the success critic does not actually correspond to any real probability; it is simply a metric corresponding to the competency of the agent. Furthermore, switching with probability based solely on the values output by success critic can lead to excessive switching. If the agent is in an area of the state-goal space where $F(s, g) \approx 0.1$, the expected number of steps the agent will take before switching goals is only 10. If the agent is in a region with even moderately high success critic values for both forward and reset goals, it could get stuck switching back and forth in that region and not explore anywhere else. The switching behavior also alters the distribution of states that the agent samples when learning, shifting it further from the state distribution of the current policy, which can lead to unstable learning.

To counteract the excessive switching behavior, we introduce three mechanisms to modulate the frequency of switching: **(1) Switching on Portion of Trajectories:** Inspired by how $\epsilon$-greedy policies only explore for a certain portion of their actions, we only apply our intelligent switching to any given forward or backward trajectory with probability $\zeta$. This decision is made at the start of the trajectory. **(2) Minimum Trajectory Length:** For the trajectories we do early switching for, we wait until the trajectory has reached a minimum length $m$ before making switching decisions. **(3) Conservative Factor:** Finally, we define the probability of switching as a function of $F(s, g)$ and trajectory length, that increases with trajectory length. Specifically, $P(switch) = F(s, g) \times (1 - \beta^t)$ where $t$ is the number of timesteps in the current trajectory and $\beta \in [0, 1]$ is a parameter that decides how conservative to be with the switching. As $t \to \infty, P(switch) \to F(s, g)$.

### 4.3 ALGORITHM SUMMARY

The early switching part of our method is summarized in Algorithms 2 and 1. At the start of a trajectory, our method decides whether to perform early switching checks for the current trajectory. Switching checks are only performed if the trajectory reaches some minimum length. Finally, the probability of switching is computed as a function of the success critic that increases with trajectory length. The agent samples a bernoulli random variable with this probability to decide whether to switch goals.

We also do timeout-nonterminal bootstrapping as described in Section 4.1 when training our agent's critic and success critic.

## 5 EXPERIMENTS

In this section, we empirically analyze the performance of RISC. Specifically, we: (1) Investigate whether reverse curriculums are the best approach for reset-free RL; (2) Compare the performance of RISC to other reset-free methods on the EARL benchmark; (3) Evaluate the necessity of both; timeout-nonterminal bootstrapping and early switching for RISC with an ablation study.

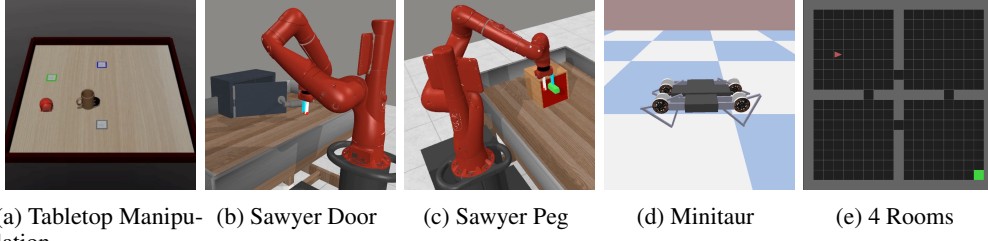

(a) Tabletop Manipu-   (b) Sawyer Door   (c) Sawyer Peg   (d) Minitaur   (e) 4 Rooms
lation

Figure 2: The environments used in Section 5. The first 4 are part of the EARL benchmark (Sharma et al., 2021b), and the last is based on Minigrid (Chevalier-Boisvert et al., 2018).

**Environments.** The experiments in Section 5.1 use a *4 rooms* gridworld where the agent needs to go from one corner to the opposite corner. The representation consists of 3 binary image channels, corresponding to the agent, the walls, and the goal location, while each cell corresponds to a spatial location. Sections 5.2 and 5.3 use four environments from the EARL benchmark (Sharma et al., 2021b): the *Tabletop Manipulation* environment (Sharma et al., 2021a) involves moving a mug to one of four locations with a gripper; the *Sawyer Door* environment (Yu et al., 2019) has a sawyer robot learn to close a door; the *Sawyer Peg* environment (Yu et al., 2019) has a sawyer robot learning to insert a peg into a goal location; the *Minitaur* environment (Coumans & Bai, 2016) is a locomotion task where a minitaur robot learns to navigate to a set of goal locations. The first 3 environments are sparse reward tasks where a set of forward and resetting demonstrations are provided to the agent, while the last environment is a dense reward environment with no demonstrations provided.

These environments all provide a low dimensional state representation for the agent. We follow the evaluation protocol outlined by Sharma et al. (2021b), evaluating the agent for 10 episodes every 10,000 steps. Every agent was run for 5 seeds. For all environments, we assume access to a reward function and a success function that can be queried for goals in the task goal distribution or initial state distribution.

**Comparisons.** We compare against the following baselines and methods: (1) Forward Backward RL (FBRL) (Eysenbach et al., 2017; Han et al., 2015), where the agent alternates between going forward to the goal and resetting back to the initial state distribution; (2) R3L (Zhu et al., 2020), which uses a novelty based reset controller; (3) VapRL (Sharma et al., 2021a), where the reset controller builds a curriculum using states in demonstration data for the forward agent based on the forward agent's performance; (4) MEDAL (Sharma et al., 2022), where the reset controller learns to reset to states in the demonstration data; (5) Naive RL, where the agent only optimizes for the task reward throughout training i.e., an episodic agent being trained in a reset-free environment; (6) Episodic RL, an agent run in an episodic environment with frequent automatic resets and is meant as an unrealistic *oracle* to understand the best possible performance. The performances for these methods were either sourced from publicly available numbers (Sharma et al., 2021b) or recreated with public implementations.

For the EARL experiments, all agents use a SAC agent (Haarnoja et al., 2018) as the base agent, with similar hyperparameters to earlier works. For the *4 rooms* experiments, all agents use a DQN (Mnih et al., 2015) agent as their base.

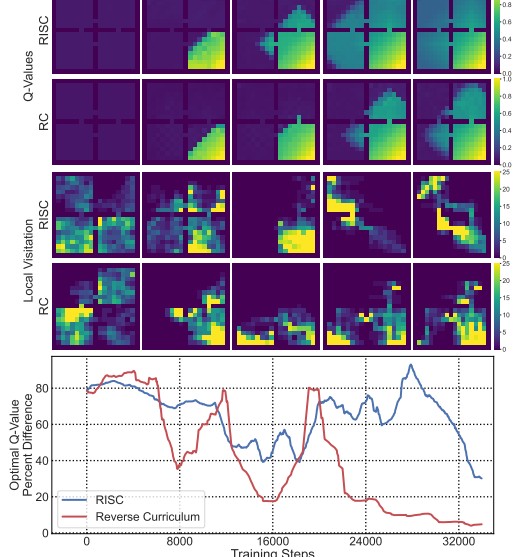

Figure 3: The heatmaps show the progression of the Q-values (top) and the visitation frequencies (middle) of the agent in forward mode in the 2000 steps after the noted timestep. The reverse curriculum method tends to be biased towards states that it has already learned, while RISC is more evenly distributed, and slightly biased towards states it hasn't fully learned.

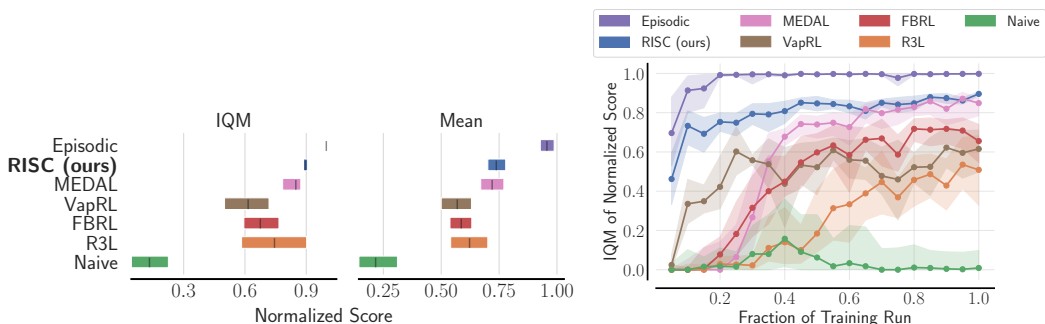

Figure 4: 95% confidence intervals for the interquartile mean (IQM) and mean normalized performance of RISC and other baselines aggregated across environments on the EARL benchmark (*Tabletop Manipulation*, *Sawyer Door*, *Sawyer Peg*, and *Minitaur*). Because MEDAL and VapRL require demonstrations and thus do not work on *Minitaur*, we exclude *Minitaur* from their calculations (left). IQM of RISC and other baselines on EARL benchmark as a function of progress through the training run. Shaded regions represent 95% confidence intervals (right). RISC outperforms and learns much faster than other reset-free baselines.

## 5.1 CURRICULA IN RESET-FREE RL

We first investigate the performance of RISC on a *4 rooms* minigrid environment, comparing it to a simple Reverse Curriculum (RC) agent. The RC agent maintains a success critic similar to RISC. When going backward, it switches to the forward controller if the success critic value falls below a given threshold. While this is not the same as other reset-free reverse curriculum methods in the literature, it is simple to implement, does not require mechanisms such as demonstrations or reward functions for arbitrary states, and is a useful tool to see differences in behavior of the agents.

Figure 3 tracks the behavior and internals of an RISC and a RC agent across time. It shows, for the noted timesteps (1) the Q-values of each policy (Figure 3, top), (2) the visitation frequency of both agents for the 2000 steps following the timestep (Figure 3, middle) (3) the difference in the agent's state value estimate and the optimal state value estimate for the experience that each agent gathers (Figure 3, bottom). We call this the Optimal Value Percent Difference (OVPD), and it is calculated as $OVPD(s) = \frac{|max_a Q_\theta(s,a) - V^*(s)|}{V^*(s)}$. We use it as a proxy for how useful the state is for the agent to collect.

The figure shows that the RISC agent tends to gather experience in areas that it has not yet mastered (i.e., Q-values are low), while the RC agent gathers much more experience in areas that it has already mastered. This is supported both by comparing the local visitation frequencies of the agents to the Q-values, and the OVPD values. The OVPD of the RC agent drops very low even before it has converged, meaning that the RC agent is not collecting transitions that it has inaccurate estimates for. Meanwhile, the RISC agent's OVPD stays relatively high throughout training, until it converges. This analysis suggests that using a reverse curriculum might not be the best approach for Reset-Free RL.

## 5.2 EVALUATING RISC ON THE EARL BENCHMARK

Figures 4 and 5 show the results of RISC on the EARL benchmark environments compared to previously published results. The aggregate performance of RISC on the EARL benchmark is better than any other reset-free method. RISC also learns the fastest both in aggregate and across multiple individual environment in the benchmark. In the scope of reset-free RL, it is fairly expensive to collect data, so any method that can reach the better policy faster is of great value.

It is also important to note the requirements and complexity of some of these methods. Both MEDAL and VapRL require demonstrations, *meaning that they cannot be run on environments such as Minitaur*. VapRL also requires access to a reward function that can be queried for goal states in the demonstration data. RISC uses demonstration data to seed the replay buffer, but does not require demonstrations to be provided. When comparing RISC to the methods that can run on all environments, RISC clearly outperfoms them all.

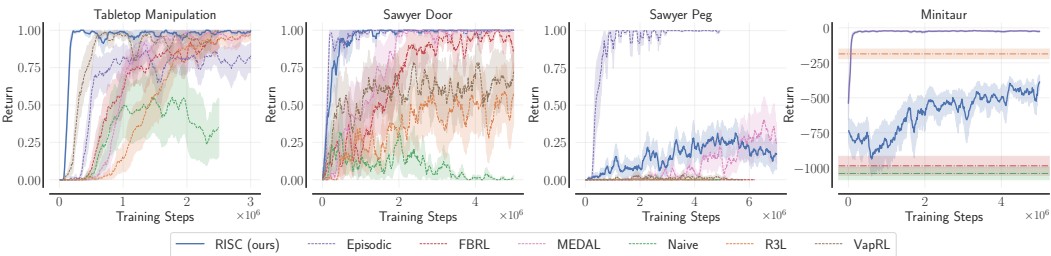

Figure 5: Average test returns on EARL benchmark tasks over timestep. RISC improves upon or matches the state-of-the-art for reset-free algorithms on 3 of the 4 environments (*Tabletop Manipulation*, *Sawyer Door*, *Sawyer Peg*), and even outperforms/matches the Episodic baseline on *Sawyer Door* and *Sawyer Peg*. Learning curves and code were not available for several baselines for *Minitaur*, so only the final performance is plotted. Results are averaged over 5 seeds and the shaded regions represent standard error.

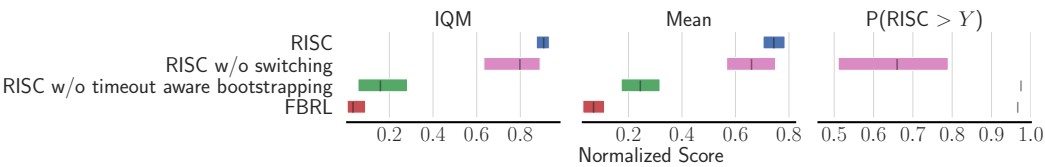

Figure 6: 95% confidence intervals for the interquartile mean (IQM), mean, and Probability of Improvement of RISC for ablations of RISC.

## 5.3 ABLATING THE COMPONENTS OF RISC

We now ablate the components of RISC to better understand the contributions of each. We rerun RISC with early switching removed and with the timeout aware bootstrapping removed. Removing both components results in a method that is equivalent to FBRL, and so we also include that in our comparison. From Figure 6, we see that removing the timeout aware bootstrapping results in a significant drop in performance, and removing early switching also results in a modest drop. This suggests that both components are important for the success of RISC.

These results are also consistent in the context of our analysis in Section 4.1. Our arguments there imply that not doing timeout aware bootstrapping can be a large issue for reset-free methods, and especially methods such as ours where the trajectory lengths can be significantly shorter than the episode lengths for the deployment task.

## 6 CONCLUSION AND FUTURE WORK

We propose a method that learns to intelligently switch between controllers in reset-free RL. It does so by (1) careful management of bootstrapping of states when switching controllers and (2) learning when to actually switch between the controllers. Our method results in performance that sets or matches the state-of-the-art on 3 of the 4 environments from the recently proposed EARL benchmark. We show success across both sparse reward tasks with demonstrations and dense reward tasks with no demonstrations. Finally, we note that at least two of the environments, *Sawyer Door* and *Tabletop Manipulation*, seem close to being saturated in terms of performance, and future work should explore other environments or ways of increasing the difficulty of those environments.

One limitation of our method is that it may not be well suited for environments with irreversible states, where the agent could get stuck. RISC tries to take advantage of the reset-free RL setting by exploring the state space more aggressively. While this is not an issue with the environments we study in this work, as they do not contain any irreversible states, it likely could be problematic in environments where there are irreversible states and safety is a concern. We also do not leverage demonstrations to guide our agent intelligently as previous works do, and learning to do so could be an interesting avenue for future work.

## ACKNOWLEDGEMENTS

Sarath Chandar is supported by the Canada CIFAR AI Chairs program, the Canada Research Chair in Lifelong Machine Learning, and the NSERC Discovery Grant. Glen Berseth acknowledges funding support from the Canada CIFAR AI Chairs program and NSERC. Janarthanan Rajendran acknowledges the support of the IVADO postdoctoral fellowship. The authors would like to thank Nishanth Anand, Mohammad Reza Samsami, and anonymous reviewers for their helpful feedback and discussions. We would also like to acknowledge the material support of the Digital Research Alliance of Canada (alliancecan.ca), Mila IDT (mila.quebec), and NVidia in the form of computational resources.

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

## A    AGENT DETAILS

All of our agents for the experiments on the EARL benchmark (Sharma et al., 2021b) use SAC (Haarnoja et al., 2018) as the base agent. The neural network architecture used for the SAC agent was taken from the publicly available implementation of the MEDAL algorithm (Sharma et al., 2022). A Linear Layer (50 units), LayerNorm, and Tanh activation are applied to the observation before being fed to the actor or critic networks. The other hyperparameters used for the base agent are described in Table 1. The experiments on the 4-rooms gridworld (Chevalier-Boisvert et al., 2018) use DQN (Mnih et al., 2015) as the base agent. The corresponding hyperparameters for those experiments are shown in Tables 2. The additional hyperparameters for RISC are shown in Table 3.

For the environments that provide demonstrations (listed in B), we simply insert those demonstrations into the replay buffer of the agents at the start of training.

| Hyperparameter | Value |
| --- | --- |
| Actor Network | MLP(256, 256) |
| Critic Network | MLP(256, 256) |
| Optimizer | Adam |
| Actor LR | 3e-4 |
| Critic LR | 3e-4 |
| $\alpha$ LR | 3e-4 |
| Weight Initialization | Xavier Uniform |
| Bias Initialization | Fill(0) |
| Target entropy scale | 0.5 |
| Reward Scale Factor | 10 |
| Critic Loss Weight | 0.5 |
| Target update $\tau$ | 0.005 |
| Target update frequency | 1 |
| Discount Factor | 0.99 |
| Batch Size | 256 |
| Initial Collect Steps | 10,000 |
| Replay Capacity | 10,000,000 |
| Collect Steps per Update | 1 |
| Min Log Std | -20 |
| Max Log Std | 10 |
| Log Std Clamp function | Clip |

Table 1: Hyperparameters for base SAC Agent

| Hyperparameter | Value |
| --- | --- |
| Q-Network | Conv([16,16,16], 3), FC(64) |
| Optimizer | Adam |
| Q-Network LR | 1e-3 |
| Target hard update frequency | 500 |
| Discount Factor | 0.95 |
| Batch Size | 128 |
| Initial Collect Steps | 512 |
| Replay Capacity | 50,000 |
| Collect Steps per Update | 1 |
| $\epsilon$-greedy init value | 1.0 |
| $\epsilon$-greedy end value | 0.1 |
| $\epsilon$-greedy decay steps | 10,000 |

Table 2: Hyperparameters for base DQN Agent

## B    EXPERIMENTAL SETUP

All experiments were run as CPU jobs. Table 4 gives further details about the experimental setup for each environment. Each configuration is run for 5 seeds. Hyperparameter search was done over the following hyperparameters: conservative factor $\beta \in \{0.0, 0.9, 0.95\}$, minimum trajectory length as a fraction of the maximum trajectory length $\in \{0.0, 0.25, 0.5, 0.75\}$, and switching trajectories proportion $\zeta \in \{0.25, 0.5, 0.75, 1.0\}$.

## C    METRICS COMPUTATION

The metrics in Figures 4 and 6 were all calculated using the rliable library (Agarwal et al., 2021). Each metric was calculated using 2000 bootstrap replications over normalized and aggregated results

---
*As a fraction of the maximum trajectory length

| | Hyperparameter | Value |
|---|---|---|
| **Tabletop Manipulation** | Conservative Factor $\beta$ | 0.9 |
| | Minimum Trajectory Length* | 0.5 |
| | Trajectories Proportion $\zeta$ | 1.0 |
| | Success Critic Network | MLP(256, 256) |
| | Success Critic Optimizer | Adam |
| | Success Critic LR | 3e-4 |
| | Success Critic Output Activation | $-0.5cos(x) - 1$ |
| | # Actions Sampled | 5 |
| **Sawyer Door** | Conservative Factor $\beta$ | 0.9 |
| | Minimum Trajectory Length* | 0.75 |
| | Trajectories Proportion $\zeta$ | 1.0 |
| | Success Critic Network | MLP(256, 256) |
| | Success Critic Optimizer | Adam |
| | Success Critic LR | 3e-4 |
| | Success Critic Output Activation | $-0.5cos(x) - 1$ |
| | # Actions Sampled | 5 |
| **Sawyer Peg** | Conservative Factor $\beta$ | 0.9 |
| | Minimum Trajectory Length* | 0.5 |
| | Trajectories Proportion $\zeta$ | 0.5 |
| | Success Critic Network | MLP(256, 256) |
| | Success Critic Optimizer | Adam |
| | Success Critic LR | 3e-4 |
| | Success Critic Output Activation | $-0.5cos(x) - 1$ |
| | # Actions Sampled | 5 |
| **Minitaur** | Conservative Factor $\beta$ | 0.95 |
| | Minimum Trajectory Length* | 0.25 |
| | Trajectories Proportion $\zeta$ | 0.25 |
| | Success Critic Network | MLP(256, 256) |
| | Success Critic Optimizer | Adam |
| | Success Critic LR | 3e-4 |
| | Success Critic Output Activation | $-0.5cos(x) - 1$ |
| | # Actions Sampled | 5 |
| **4 Rooms** | Conservative Factor $\beta$ | 0.95 |
| | Minimum Trajectory Length* | 0 |
| | Trajectories Proportion $\zeta$ | 0.5 |
| | Success Critic Network | Conv([16,16,16], 3), FC(64) |
| | Success Critic Optimizer | Adam |
| | Success Critic Output Activation | $sigmoid(x)$ |
| | Success Critic LR | 1e-3 |

Table 3: Hyperparameters for RISC

| | # Train Steps | # Train Hours | Hard Reset Frequency | Epsisode length limit | # of Demo Transitions | Reward Type |
|---|---|---|---|---|---|---|
| Tabletop Manipulation | 3,000,000 | 24 | 200,000 | 200 | 2,534 | Sparse |
| Sawyer Door | 5,000,000 | 42 | 200,000 | 300 | 1,095 | Sparse |
| Sawyer Peg | 7,000,000 | 68 | 100,000 | 200 | 1,815 | Sparse |
| Minitaur | 5,000,000 | 50 | 100,000 | 1,000 | 0 | Dense |
| 4-Rooms | 50,000 | 0.5 | 50,000 | 100 | 0 | Sparse |

Table 4: Experimental setup details for each environment.

from all tasks. For *Minitaur*, we did not have the run data for all of the baselines, only the final mean and standard error. To overcome this and integrate *Minitaur* results in our comparisons, we generated 1000 sets of samples from the distribution represented by the mean and standard error of

|  | Tabletop Organization | Sawyer Door | Sawyer Peg | Minitaur |
|---|---|---|---|---|
| *Naive RL* | $0.32 \pm 0.17$ | $0.00 \pm 0.00$ | $0.00 \pm 0.00$ | $-1041.10 \pm 44.58$ |
| *R3L* | $0.96 \pm 0.04$ | $0.54 \pm 0.18$ | $0.00 \pm 0.00$ | $-186.30 \pm 34.79$ |
| *VaPRL* | $0.98 \pm 0.02$ | $0.94 \pm 0.05$ | $0.02 \pm 0.02$ | - |
| *MEDAL* | $0.98 \pm 0.02$ | $\mathbf{1.00 \pm 0.00}$ | $\mathbf{0.40 \pm 0.16}$ | - |
| *FBRL* | $0.94 \pm 0.04$ | $\mathbf{1.00 \pm 0.00}$ | $0.00 \pm 0.00$ | $-986.34 \pm 67.95$ |
| *RISC (ours)* | $\mathbf{1.0 \pm 0.0}$ | $\mathbf{1.0 \pm 0.0}$ | $\mathbf{0.42 \pm 0.07}$ | $-321.74 \pm 53.73$ |
| *Oracle RL* | $0.80 \pm 0.11$ | $1.00 \pm 0.00$ | $1.00 \pm 0.00$ | $-41.50 \pm 3.40$ |

Table 5: Average return of the best policy over 5 random seeds, reported with standard error. Higher returns are better, and the best method for each environment (excluding oracle) is in bold. For *Tabletop Manipulation*, *Sawyer Door*, and *Sawyer Peg*, minimum performance is 0.0 and maximum performance is 1.0.

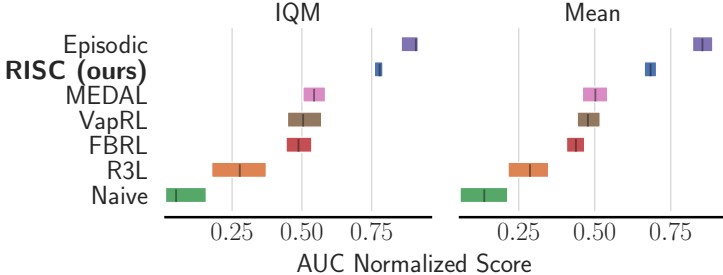

Figure 7: Interquartile mean and mean for the area under the curve (AUC) for the learning curve of each run, aggregated across runs and environments. Higher AUC is better and implies more sample efficient learning.

each baseline, computed rliable metrics with each set, and used the set with the highest mean, to give as much benefit of the doubt as possible to the baselines. Despite this bias, the difference between the best set and the worst set was quite small, with the largest difference being approximately .05 for FBRL.

## D  EARL RESULTS

We present the results for the best policy of each method on the environments in the EARL benchmark for direct comparison with previous methods in Table 5. Note, RISC has the best performance of reset free methods on 3 of 4 environments, and performs somewhat competitively on the last environment.

We also present the interquartile mean and mean for the area under the curve (AUC) of the learning curves for each agent, aggregated across seeds and environments (Figure 7). RISC is significantly higher than the other methods, implying that it is more sample efficient.

## E  EXPLORING THE MODULATING MECHANISMS

In this section, we show the necessity of the modulations introduced in Section 4.2.2. In Figure 9, we see that removing the modulations significantly reduces the average trajectory length of the agent. Furthermore, when we compare the performance of the RISC with and without modulations in Figure 8, we see that without the modulations, RISC underperforms on all environments.

## F  HYPERPARAMETER ANALYSIS

We first present details of our hyperparameter sensitivity analysis. We took the optimal configurations selected for each environment, and varied the value of one hyperparameter at a time. The results in

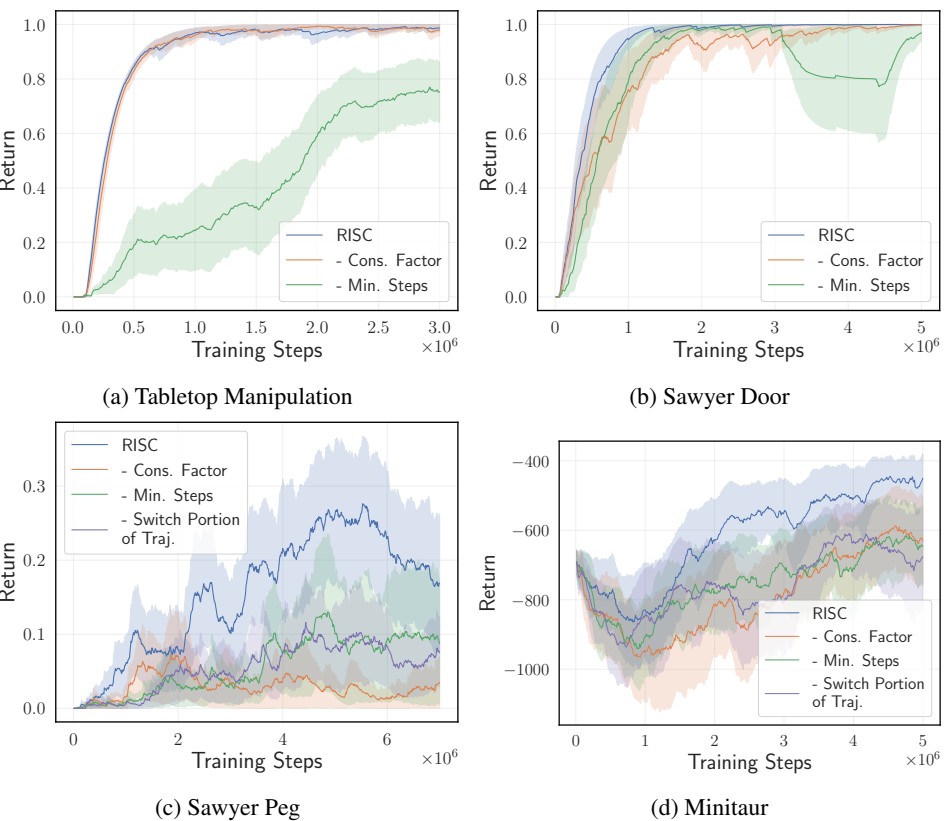

(a) Tabletop Manipulation

(b) Sawyer Door

(c) Sawyer Peg

(d) Minitaur

Figure 8: We show the effect of removing the modulations presented in Section 4.2.2. The performance of the agent degrades across all environments, with significant deterioration on *Minitaur* and *Sawyer Peg*.

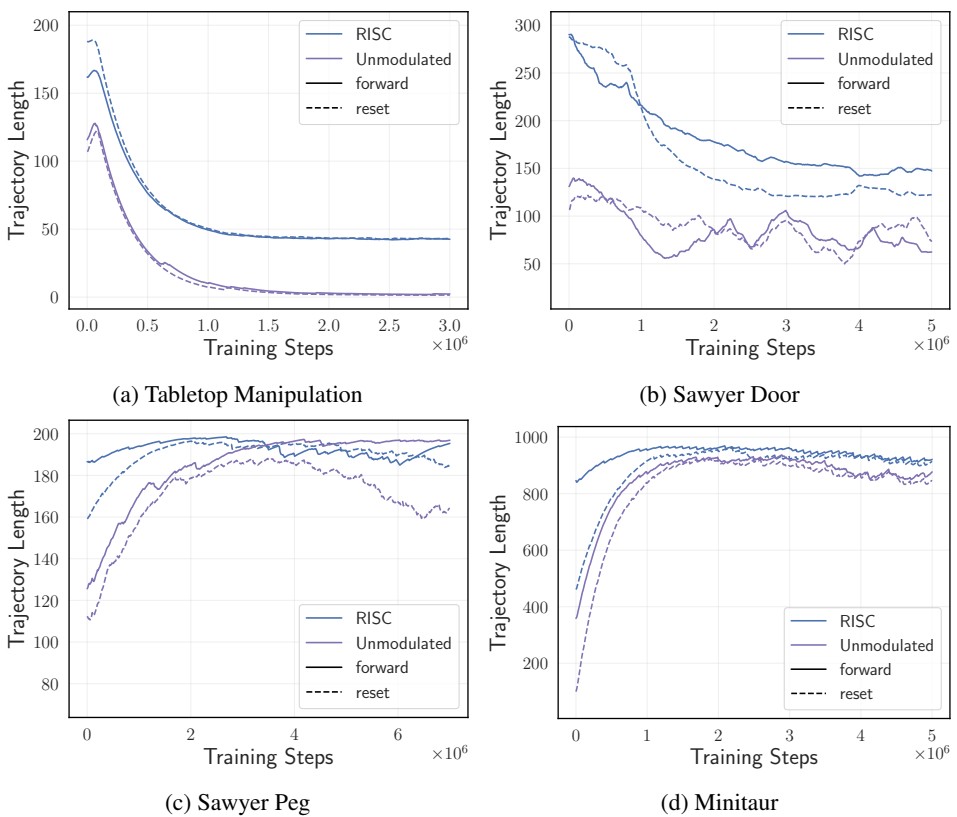

(a) Tabletop Manipulation

(b) Sawyer Door

(c) Sawyer Peg

(d) Minitaur

Figure 9: We show the effect of removing the modulations presented in Section 4.2.2 on the trajectory lengths of the agent. As expected, the average trajectory length decreases significantly on all environments.

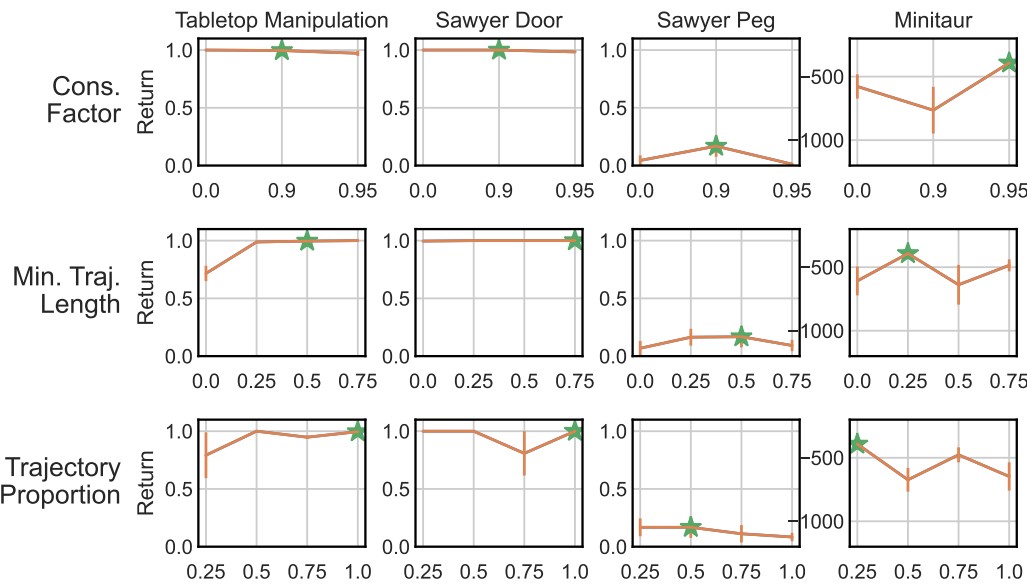

Figure 10: Hyperparameter sensitivity analysis. We vary the selected configuration one hyperparameter at a time, and present the mean and standard deviation for each configuration. The configurations used in the experiments are marked with a green star.

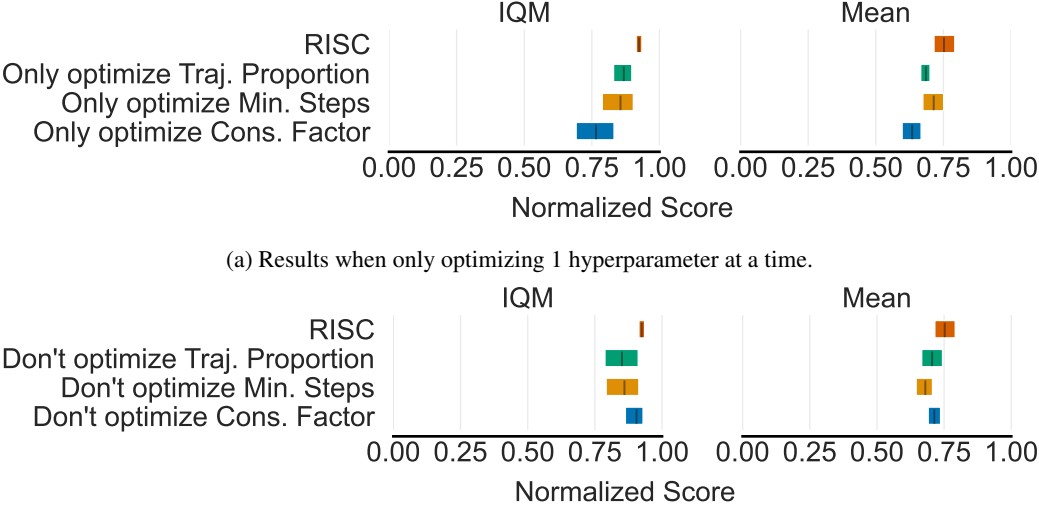

(a) Results when only optimizing 1 hyperparameter at a time.

(b) Results when only optimizing 2 hyperparameters at a time.

Figure 11: Results of partial hyperparameter optimization.

Figure 10 show that our results are fairly robust to changes in hyperparameters, with fairly small changes to the final return across all environments.

For our second analysis, we tried optimizing only 1 hyperparameter at a time. For each environment, we select the configuration with the best performance for each hyperparameter where the other modulations are turned off, and aggregate across environments. We do a similar analysis with optimizing 2 hyperparameters at a time. The results are shown in Figure 11a and Figure 11b respectively. The most important hyperparameters seem to be the minimum trajectory length or the proportion of trajectories where the switching happens. They recover most of the performance, but the full method still results in a slight performance gain.

