# OpenReview forum: "Intelligent Switching for Reset-Free RL"
_ICLR.cc/2024/Conference — ICLR 2024 poster_

### Official Review · Reviewer_VmY7 · 2023-10-24

**Soundness:** 2 fair
**Presentation:** 3 good
**Contribution:** 2 fair
**Rating:** 8
**Confidence:** 3

**Summary:**

The paper presents a novel approach to tackle the challenge of training reinforcement learning agents in real-world scenarios without relying on strong episode resetting mechanisms. The authors propose a method called Reset Free RL with Intelligently Switching Controller (RISC) that learns to switch between controllers intelligently. RISC addresses the crucial aspects of managing state bootstrapping during controller transitions and determining when to switch between controllers. The experimental results demonstrate that the proposed method achieves state-of-the-art performance on various challenging environments for reset-free RL, including sparse and dense reward tasks, though the authors suggest further exploration of new environments or enhanced difficulty levels. However, a limitation of RISC is its potential unsuitability for environments with irreversible states, where safety concerns may arise. Additionally, the paper acknowledges the absence of leveraging demonstrations for agent guidance, which could be an interesting direction for future research.

**Strengths:**

The paper's key strength lies in its innovative approach to addressing the underexplored aspect of learning when switching between controllers in reset-free reinforcement learning. By recognizing the absence of an imposed episode time limit in this setting and considering the duration of controller trajectories as a parameter, the authors introduce a novel concept: dynamically learning when to switch controllers based on the agent's ability to achieve its current goal. This intelligent switching mechanism not only optimizes the agent's experience collection but also facilitates more efficient learning by focusing on unmastered states and goals. The proposed algorithm, Reset Free RL with Intelligently Switching Controller (RISC), is evaluated on a challenging benchmark of robot manipulation and navigation tasks, demonstrating state-of-the-art performance in various reset-free environments. This novel approach to controller switching has the potential to significantly enhance the effectiveness of reset-free reinforcement learning algorithms, addressing an important and previously unexplored aspect of the field.

**Weaknesses:**

- The paper has no significant theoretical results.
- The convergence of the proposed method is not guaranteed.
- Too many hyperparameters to tune, e.g., minimum trajectory length, conservative factor...

**Questions:**

- In the last paragraph of 4.2.1, could the author explain more about why we should set $\gamma_{sc}$ to be the value of the critic's discount factor?
- Could the author explain more about how the method makes the learning stable?
- I suppose the work will benefit the reward-free RL (Yang, Q., & Spaan, M. T. (2023). CEM: Constrained Entropy Maximization for Task-Agnostic Safe Exploration. AAAI-2023). Could the author elaborate a bit on how to use the ideas in reward-free RL for exploration?

---

> ### Author Response · Authors · 2023-11-17
> **Author Response**
>
> We thank the reviewer for their comments. We are glad the reviewer recognizes our innovative approach to this problem by learning when to switch. We address your concerns and questions below.
>
> **[No Significant Theoretical Results and lack of Convergence Proofs]**
> Although we do not provide a proof of the convergence properties of RISC, we do provide a discussion of the role of bootstrapping in the reset-free RL setting in Section 4.1. Reset-Free RL thus far has been a fairly empirical field, with several prior works [1,2,3,4] being accepted to ICLR and comparable conferences without theoretical results. While it would be useful, developing such theory falls outside the scope of this work.
>
> **[Too many hyperparameters]**
> We agree that adding hyperparameters adds complexity to the method, but we would like to point out: (1) Our algorithm is fairly robust to hyperparameter changes, and our performance does not degrade drastically even when completely removing a hyperparameter from the method. To demonstrate this, we have added a hyperparameter sensitivity analysis in Figure 10 (Appendix F, new figure) and Figure 11 (Appendix F, new figure) showing the results when only optimizing for 1 or 2 of the 3 parameters. In both cases, the perturbation/removal of hyperparameters reduced performance, but not significantly. (2) Having those hyperparameters is necessary to maintain the flexibility of our method. Unlike our work, several prior methods only work on sparse reward environments with demonstrations, and the ones that also work on dense reward environments without demonstrations, we outperform. Thus, for the cost of the extra hyperparameters, we provide added coverage on different environments.
>
> **[Value of success critic's discount factor]**
> This was simply a design choice based on the fact that we wanted the success critic to generally reflect the capabilities and state evaluation of the main agent. A different discount factor could be used, and could potentially result in more or less conservative switching behavior.
>
> **[How the method makes learning more stable]**
> The timeout-nonterminal bootstrapping strategy makes learning more stable, as it computes consistent targets for a given state regardless of whether a trajectory ended there. With the other strategy, if a transition is sampled from the replay, and it represents the end of the trajectory but not a terminal state, it still gets assigned a target value of 0. If this transition also exists in the replay buffer as not the end of a trajectory, it gets assigned a target value based on the next state in the transition. Removing this discrepancy by using the timeout-nonterminal bootstrapping strategy leads to more stable learning.
>
> **[Relation to CEM and Reward-Free RL]**
> The ideas from CEM and Reward-Free RL seem complementary to the ideas of our work and potentially useful in adapting Reset-Free RL methods to unsafe environments with irreversible states. Our method allows the agent to explore areas it hasn’t mastered yet, and the ideas from CEM can be used to guide the agent and perform safe exploration.
>
> We thank the reviewer for their valuable suggestions and helping improve our work. If your major questions and concerns have been addressed, we would appreciate it if you could further support our work by increasing your score. If there are more questions/concerns, please let us know.

---

> > ### Comment · Reviewer_VmY7 · 2023-11-18
> >
> > Thanks to the authors for their response. In general, most of my concerns were addressed. But I have two more questions:
> >
> > Authors' response: **How the method makes learning more stable**
> >
> > I agree the timeout-nonterminal bootstrapping strategy could make learning more stable. As to the traditional RL algorithms, how timeout-nonterminal bootstrapping strategy will benefit them? What are the obstacles for the traditional RL to use the timeout-nonterminal bootstrapping strategy?
> >
> > I noticed that the authors did not indicate whether they will open-source their code. What are the considerations on this point?

---

> ### Author Response · Authors · 2023-11-18
>
> Thank you for your response and for engaging in the discussion process. We respond to your questions below.
>
> **[Timeout-nonterminal bootstrapping for traditional RL]**
> Timeout-nonterminal bootstrapping has shown benefits for traditional RL as well in terms of sample efficiency [1]. There is no major obstacle for traditional RL to use timeout-nonterminal bootstrapping other than awareness and getting the community to adopt them. The new versions of gym (Gymnasium [2]) and dm_env [3] have moved to an API that more clearly supports the distinction between termination and truncation. The challenge is now to get the community to update legacy environment code to match these APIs and legacy agent code to use the new API.
>
> **[Open-sourcing code]**
> Yes, we will release code with the final version of the paper.
>
> If we have addressed your questions and concerns, we would appreciate it if you could support our work by raising your score. As before, if you have any more questions/concerns, please let us know.
>
> [1] Pardo, Fabio, et al. “Time limits in reinforcement learning.” International Conference on Machine Learning. PMLR, 2018.
> [2] Towers, Mark, et al. “Gymnasium.” 2023.
> [3] Alistair Muldal, et al. “dm_env: A Python interface for reinforcement learning environments.” 2019.

---

> > ### Comment · Reviewer_VmY7 · 2023-11-19
> >
> > Thanks for the reply. I am happy to raise my score.

---

### Official Review · Reviewer_gA9F · 2023-10-30

**Soundness:** 3 good
**Presentation:** 3 good
**Contribution:** 3 good
**Rating:** 8
**Confidence:** 4

**Summary:**

Summary:
The paper addresses the reset-free reinforcement learning (RL) setting, which assumes the absence of a reset mechanism, making it a more realistic scenario due to the potential cost associated with resets in the real world. Existing RL methods typically train two separate agents: one toward the goal and another toward the initial state, aiming to bring the agent back to the initial state for additional practice. This paper contends that intelligently switching between these two agents, particularly when the agent is competent, is crucial. Furthermore, the addition of bootstrapping for the last state before switching is shown to significantly enhance performance. The proposed method, referred to as RISC, outperforms state-of-the-art reset-free RL methods on four EARL tasks (a common reset-free RL benchmark) and a four-room navigation task.

**Strengths:**

1. The paper is well-written, experiments are well-executed, and details of implementations are reported.

2. The paper raises a significant concern regarding the handling of time-out non-terminal states in RL, which is often overlooked. It emphasizes the importance of correctly handling bootstrapping particularly in the reset-free context.

3. The concept of intelligently switching between two different agents is intriguing, opening the door to further research in this direction.

**Weaknesses:**

1. One important argument this paper makes is that adding bootstrapping for time-out non-terminal states is important, although it is theoretically well motivated, I think it would be better to see some practical motivations, especially why it is important under the reset-free setting. For example, are value estimations very different?

2. The paper introduces several method-specific hyper-parameters such as M, m, and β. It would be valuable to discuss the method's sensitivity and robustness to these hyper-parameters.

**Questions:**

1. Will code be available?

2. Are there any correlations between adding bootstrapping and adding an intelligent switch? For example, will switch be **more** useful when bootstrapping is added?

3. If I understand correctly, a separate critic success function was trained. Why didn't the authors use the critic from SAC (Soft Actor-Critic) instead of training a new one?

4. In EARL paper, they claimed not using bootstrapping can break the long TD chain, in contrast, this paper suggests using bootstrap. Could authors also discuss these two different ideas?

5. Bootstrapping for time-out non-terminal states should always be performed, do you have any intuitions on is it more important in the reset-free setting, since in episodic RL setting, people generally just ignore it?

6. At the beginning of section 5.1, the paper discusses RC and RISC. Could you please elaborate more on RC and the difference between these two agents? Does RC also use bootstrapping?

7. Analysis on Fig.3 (the last paragraph of section 5.1), you mentioned the RISC agent tends to visit areas where Q-values are low. But in the third/fourth column of Fig.3, it seems like the RC agent also tends to visit areas where Q-values are low? In the third column, the RISC agent actually gathers lots of data in the first room where Q-values are already quite high.

8. There’s a performance drop on sawyer peg tasks, do you have any intuitions on the decrease of performance?

9. Which task do you perform ablation study on in Fig.6?

10. You mentioned demo data is put into the replay buffer, do you oversample it? Will RISC perform the same without demo data?

11. There's a recent work called IBC (Demonstration-free Autonomous Reinforcement Learning via Implicit and Bidirectional Curriculum) from ICML 2023. While it is relatively new, it may be worth discussing it in the related work section without the need for experimental comparisons.

---

> ### Author Response · Authors · 2023-11-17
> **Author Response (1/2)**
>
> We thank the reviewer for their insightful comments. We’re glad they found our paper well written, our experiments well executed, our analysis of the bootstrapping strategies valuable, and the idea behind our intelligent switching mechanism to be intriguing.  We address your concerns and questions below.
>
> **[Practical motivations for timeout non-terminal bootstrapping]**
> The timeout-nonterminal bootstrapping strategy makes learning more stable, as it computes consistent targets for a given state regardless of whether a trajectory ended there. With the other strategy, if a transition is sampled from the replay, and it represents the end of the trajectory but not a terminal state, it still gets assigned a target value of 0. If this transition also exists in the replay buffer as not the end of a trajectory, it gets assigned a target value based on the next state in the transition. Removing this discrepancy by using the timeout-nonterminal bootstrapping strategy leads to more stable learning.
>
> **[Hyperparameter Sensitivity Analysis]**
> Our algorithm is fairly robust to hyperparameter changes, and our performance does not degrade drastically when completely removing a hyperparameter from the method. To demonstrate this, we have added a hyperparameter sensitivity analysis in Figure 10 (Appendix F, new figure), and Figure 11 (Appendix F, new figure) showing the results when only optimizing for 1 or 2 of the 3 parameters. In both cases, the perturbation/removal of hyperparameters reduced performance, but not significantly.
>
> **[Will code be available?]**
> Yes, we will release code with the final version of the paper.
>
> **[Are there any correlations between adding bootstrapping and adding an intelligent switch]**
> The timeout-nonterminal bootstrapping strategy is necessary in order to do more aggressive switching. Since intelligent switching introduces more trajectory ends into the replay buffer, not having timeout-nonterminal bootstrapping could likely result in even more instability. This can be seen with the ablation experiments in section 5.3, where just the intelligent switching isn’t able to learn well, but adding bootstrapping and the switching results in the best agent.
>
> **[Why not use the SAC critic instead of training a separate success critic?]**
> We trained a separate success critic because the competency metric we wanted to estimate doesn’t always correspond to the discounted return that the SAC critic is estimating (e.g., when using dense rewards or reward scaling).
>
> **[Not using bootstrapping to break the long TD chain, as in EARL]**
> The EARL paper refers to breaking the TD chain for long trajectories when performing standard RL with large episode lengths. The tasks they do these experiments for are infinite horizon tasks without a notion of a task end. For such tasks, it might make sense to periodically use terminal state bootstrapping, since otherwise, trajectories for a single task can grow to be tens or hundreds of thousands of steps long. In our environments, while we are training in a reset-free setting, our target task has a notion of a task end and we get terminal signals naturally from the environment when the agent reaches the goal. Further investigating how the characteristics of the environment interact with the bootstrapping strategy is an interesting direction for future work.
>
> **[Is bootstrapping timeout-nonterminal states more important in the reset-free setting]**
> We expect timeout-nonterminal bootstrapping to likely be more important for the reset-free setting. There is both empirical and theoretical evidence for this. Based on the analysis in section 4.1, we expect that having the correct timeout-nonterminal bootstrapping strategy helps stabilize learning in the reset-free setting. Since the state distribution for a reset-free agent can be very different from the forward policy’s state distribution, stabilizing the targets through the timeout-nonterminal bootstrapping is necessary to learn. In standard RL, because the state distribution more closely matches the distribution of the forward policy, the targets will likely automatically be more stable even with the timeout-terminal strategy (although it’s been shown that doing timeout-nonterminal bootstrapping can still help). Empirically, the community has only recently started to shift to timeout-nonterminal bootstrapping for episodic tasks, and still saw much success with the old strategy. In reset-free RL, several tasks cannot be learned at all or are learned suboptimally when using the timeout-terminal strategy.

---

> > ### Author Response · Authors · 2023-11-17
> > **Author Response (2/2)**
> >
> > **[Difference between RC and RISC]**
> > RC and RISC both use bootstrapping and the same base DQN agent. The only difference is the switching mechanism. The RISC agent tries to switch when it’s entering zones where it has high Q-values, while the RC agent takes a more traditional curriculum approach of trying to switch back when encountering states it has very low Q-values for. The experiment is to demonstrate the point that ideas behind things such as reverse curriculums and the way they are used in episodic settings might not transfer well to reset-free settings.
> >
> > **[RC vs RISC analysis in Figure 3]**
> > This experiment was just a qualitative example of the trend that occurs with RC and RISC agents. Column 3 is a slight outlier in showing the RC agent having better exploration than the RISC agent, but you can also see that the level of exploration that the RC agent reaches in column 4 already happens in the RISC agent by column 2. In Column 4, the RISC agent almost exclusively explores the region of the state space that it hasn’t learned yet.
> >
> > **[Performance drop on Sawyer Peg]**
> > Near the end of training, one of our seeds for that task diverged in performance. It’s not clear why that happened.
> >
> > **[Fig. 6 Ablation tasks]**
> > We used the same tasks as Figures 4 and 5 for the ablation experiments.
> >
> > **[Use of demo data with RISC]**
> > We don’t do any special treatment of the demonstration data other than putting it into the agent’s replay buffer at the beginning of training. RISC definitely can learn without demonstration data, but it would likely be slightly worse for the sparse reward environments.
> >
> > **[Discussion of IBC]**
> > This is quite a relevant work to our paper, as they also try to create curriculums for both the forward and reset agents. Thank you for pointing it out, we will add it to our related work section.
> >
> > We thank the reviewer for their valuable comments and helping improve our work. If your major questions and concerns have been addressed, we would appreciate it if you could further support our work by increasing your score. If there are more questions/concerns, please let us know.

---

> > > ### Author Response · Authors · 2023-11-20
> > >
> > > Dear Reviewer gA9F,
> > >
> > > We hope that you've had a chance to read our responses and clarifications. As the end of the discussion period is approaching, we would greatly appreciate it if you could confirm that our updates have addressed your concerns.
> > >
> > > Thank you for your time,
> > > The Authors

---

> > > > ### Comment · Reviewer_gA9F · 2023-11-21
> > > > **Thank you for your Response**
> > > >
> > > > I think your responses addressed most of my concerns, thank you very much. I do not have further questions, and maintain my current score.

---

### Official Review · Reviewer_J5Lq · 2023-10-31

**Soundness:** 2 fair
**Presentation:** 2 fair
**Contribution:** 2 fair
**Rating:** 5
**Confidence:** 3

**Summary:**

This paper considers the problem of reset-free RL where automatic reset does not apply. Different from existing work, this paper proposes a new algorithm that switchs between forward and backward controllers intelligently. More specifically, the proposed switching function take the competent of state into account when the specific state has been explored well.  For this purpose, a Q-learning style algorithm is proposed to estimate the competency of a policy in a specific state. Through empirical experiments, the proposed algorithm is shown to achieve better performance comparing to the baselines on benchmarks.

**Strengths:**

The considered problem in this paper is interesting and has great potential in real applications as episodic RL could be hard to achieve.
In terms of different terminal strategies, this paper theoretically analyze different terminal strategies in term of bootstrapping for the final state. As timeout-terminal strategy bring more challenges to the problem, it is more recommended to have timeout-terminal loss when switching controllers. The analysis is rigorous and easy to follow.
The idea of defining and learning the competent of state and policy for preventing unnecessary exploration is cogent.

**Weaknesses:**

Although the idea of switching according to competent makes sense, I have some concerns regarding the limitations of the proposed switching function. In order to have valid result, you need additional mechanisms to modulate the frequency of switching. It could be tricky to tune \epsilon,\beta, and the minimum length. There is always a tradeoff here, as you increase the constraints, your proposed method will gain less benefits.

**Questions:**

In the paper, the authors did not explain details about the hyperparameters needed in the proposed algorithm. In 4.2.2, there is some description. But I did not see any insight on how to tune these three parameters in different scenarios. And these parameters are essential for the algorithm to work properly. I would like to read more on this.

---

> ### Author Response · Authors · 2023-11-17
> **Author Response**
>
> We thank the reviewer for their insightful comments. We’re glad they found our analysis of the bootstrapping strategies rigorous and easy to follow, and our idea of using competency of the policy to control switching to be cogent. To address your concerns, we provide a discussion of the few hyperparameters our method introduces below. The content of this discussion has also been added to Appendix F of the paper.
>
> **[Tradeoff of adding hyperparameters]**
> We agree that adding hyperparameters adds complexity to the method, but we would like to point out: (1) Our algorithm is fairly robust to hyperparameter changes (Figure 10, Appendix F, new figure), and our performance does not degrade drastically even when completely removing a hyperparameter from the method (Figure 11, Appendix F, new figure) (2) Having those hyperparameters is necessary to maintain the flexibility of our method. Unlike our work, several prior methods only work on environments with demonstrations, and we convincingly outperform the methods that can work without demonstrations. Thus, for the cost of the few extra hyperparameters, we provide added coverage on a broader set of environments.
>
> **[Hyperparameter Selection]**
> As mentioned in Appendix B, to select the hyperparameters for our experiments, we conducted a grid search over the following hyperparameters: conservative factor $\beta\in\{0.0, 0.9,0.95\}$, minimum trajectory length as a fraction of the maximum trajectory length $\in\{0.0, 0.25, 0.5, 0.75\}$, and switching trajectories proportion $\zeta\in\{0.25, 0.5, 0.75, 1.0\}$.
>
> **[Hyperparameter Sensitivity Analysis]**
> We also conducted a sensitivity analysis of each hyperparameter on each environment and added it as Figure 10 (appendix F, new figure). We took the configurations selected for each environment, and varied the value of one hyperparameter at a time. The results show that our results are fairly robust to changes in hyperparameters, with fairly small changes to the final return across all environments.
>
> **[Importance of each hyperparameter]**
> We also want to discuss the relative importance of each hyperparameter. In Figure 8 (appendix E, existing figure), we show learning curves demonstrating the effect of removing each modulation on the performance. Continuing that discussion, we add Figure 11 (appendix F, new figure) showing the effect of doing a partial hyperparameter search. Figure 11.a shows the performance achieved when only optimizing one hyperparameter, and disabling the other modulations. The most important modulations seem to be the minimum trajectory length or the proportion of trajectories where the switching happens. They recover most of the performance, but the full method still results in a slight performance gain. Figure 11.b shows the performance when optimizing 2 out of the 3 hyperparameters, and shows that each pair of hyperparameters can get fairly close to the full method’s performance, but still does slightly worse.
>
>
> We thank the reviewer for their valuable suggestions and helping improve our work. If your major questions and concerns have been addressed, we would appreciate it if you could further support our work by increasing your score. If there are more questions/concerns, please let us know.

---

> > ### Author Response · Authors · 2023-11-20
> >
> > Dear Reviewer J5Lq,
> >
> > We hope that you've had a chance to read our responses and clarification. As the end of the discussion period is approaching, we would greatly appreciate it if you could confirm that our updates have addressed your concerns.
> >
> > Thank you for your time,
> > The Authors

---

### Official Review · Reviewer_oJUJ · 2023-11-01

**Soundness:** 3 good
**Presentation:** 3 good
**Contribution:** 3 good
**Rating:** 6
**Confidence:** 3

**Summary:**

The paper presents Reset Free RL with Intelligently Switching Controller (RISC), a novel algorithm for reinforcement learning in reset-free environments. RISC intelligently switches between two agents: a forward agent that learns the task and a backward agent that resets the agent to favorable states. The key ideas are proper bootstrapping when switching controllers and learning when to switch between the agents. The authors demonstrate that RISC achieves state-of-the-art performance on several challenging environments from the EARL benchmark.

**Strengths:**

* RISC addresses the limitations of episodic RL in real-world applications, where resetting the environment is expensive and difficult to scale.
* The algorithm intelligently switches between forward and backward agents, maximizing experience generation in unexplored areas of the state space.
* RISC achieves state-of-the-art performance on several challenging environments from the EARL benchmark.

**Weaknesses:**

* The paper does not provide a thorough analysis of the theoretical properties of RISC, such as convergence guarantees.
* The experiments are limited to a small set of environments, and it is unclear how RISC would perform on more complex tasks or in other domains.

**Questions:**

* How does RISC compare to other reset-free RL algorithms in terms of sample efficiency and generalization?
* Can RISC be extended to handle environments with irreversible states, where the agent could get stuck?
* How does RISC perform when combined with other techniques, such as curriculum learning or demonstration-based learning?

---

> ### Author Response · Authors · 2023-11-17
> **Author Response**
>
> We thank the reviewer for their comments. We are glad the reviewer appreciates the strength of our results and that intelligent switching can lead to better exploration in a reset-free setting. We address the reviewers' concerns and questions below.
>
> **[The paper does not provide a thorough analysis of the theoretical properties of RISC]**: While developing such theory would be useful, Reset-Free RL thus far has been a fairly empirical field, with several prior works [1,2,3,4] being accepted to ICLR and comparable conferences without theoretical results, and developing such theory falls outside the scope of this work. Although we do not provide a proof of the convergence properties of RISC, we do provide a discussion of the role of bootstrapping in the reset-free RL setting in Section 4.1.
>
> **[Experiments limited to small set of environments]**
> We perform experiments across 3 sparse reward robotic manipulation with demonstrations (tabletop, sawyer door, sawyer peg), 1 dense reward robotic navigation without demonstrations (minitaur), and 1 sparse reward gridworld task without demonstrations (4rooms).
> Recently published works in the Reset-Free RL have a less diverse evaluation than our work, such as MEDAL [4] (tabletop, sawyer door, sawyer peg) and VapRL [2] (tabletop, sawyer door, hand manipulation), as these are all sparse reward tasks with demonstrations. In fact, we evaluate on every publicly available environment in the EARL benchmark other than Franka Kitchen, which no reset-free method has made any progress on.
>
> **[Comparison of RISC to other reset-free algorithms in terms of sample efficiency and generalization]**
> We refer the reviewer to Figures 4 and 5 (main paper, existing figures) for a summary of RISC’s performance in terms of generalization. Our algorithm has the best performance on two of the evaluated environments, the second best performance on the other two, and across all environments, has the highest mean and interquartile mean. We also have added Figure 7 (Appendix D, new figure) to highlight our method’s gains in sample efficiency compared to prior work. The figure shows the area under the curve of the learning curves, and shows that our method is significantly more sample efficient than the other baselines. Thank you for this helpful suggestion.
>
>
>
> **[Can RISC be extended to handle environments with irreversible states, where the agent could get stuck?]**
> Yes, we believe that RISC could be extended to handle environments with irreversible states. It would likely involve making the exploration more conservative, perhaps incorporating ideas from  the risk-averse RL literature [5]. An alternative approach would be to incorporate prior knowledge about the environment and unsafe states into the switching algorithm through the use of large pretrained models.
>
> **[Combining RISC with curriculum learning or demonstration-based techniques]**
> RISC by itself does form a type of implicit curriculum, as it switches directions whenever it encounters a state it has already solved, thus focusing on states it hasn’t solved. We agree with the reviewer that finding ways to combine RISC with more sophisticated curriculum learning methods or finding ways of using demonstrations more effectively is an important direction of research and could enhance the performance of our method, but this would require several more significant research contributions and is out of scope for this work.
>
> We thank the reviewer for their valuable comments and helping improve our work. If your major questions and concerns have been addressed, we would appreciate it if you could further support our work by increasing your score. If there are more questions/concerns, please let us know.
>
>
> [1] Eysenbach, Benjamin, et al. "Leave no Trace: Learning to Reset for Safe and Autonomous Reinforcement Learning." International Conference on Learning Representations. 2018.
> [2] Sharma, Archit, et al. "Autonomous reinforcement learning via subgoal curricula." Advances in Neural Information Processing Systems 34 (2021): 18474-18486.
> [3] Sharma, Archit, et al. "Autonomous Reinforcement Learning: Formalism and Benchmarking." International Conference on Learning Representations. 2021.
> [4] Sharma, Archit, Rehaan Ahmad, and Chelsea Finn. "A State-Distribution Matching Approach to Non-Episodic Reinforcement Learning." International Conference on Machine Learning. PMLR, 2022.
> [5] Brunke, Lukas, et al. "Safe learning in robotics: From learning-based control to safe reinforcement learning." Annual Review of Control, Robotics, and Autonomous Systems 5 (2022): 411-444.

---

> > ### Author Response · Authors · 2023-11-20
> >
> > Dear Reviewer oJUJ,
> >
> > We hope that you've had a chance to read our responses and clarification. As the end of the discussion period is approaching, we would greatly appreciate it if you could confirm that our updates have addressed your concerns.
> >
> > Thank you for your time,
> > The Authors

---

> > > ### Comment · Reviewer_oJUJ · 2023-11-21
> > > **Reply To The Authors**
> > >
> > > Dear Authors.
> > >
> > > Thank you for your detailed replies, the current ones solved my confusion. I will consider the comments of other reviewers and then further consider my decision.

---

### Author Response · Authors · 2023-11-17
**Global Response**

We thank all the reviewers for their comments and questions. We are glad that the reviewers appreciated our state-of-the-art results (**oJUJ, gA9F, VmY7**), the potential of our work for real applications (**oJUJ, J5Lq**), our innovative approach of learning when to switch (**oJUJ, J5Lq, gA9F, VmY7**), our analysis of bootstrapping in reset-free RL (**J5Lq, gA9F**), and our overall clarity (**J5Lq, gA9F**).

We respond to specific concerns below, and provide an updated manuscript with the following changes (highlighted in blue in the pdf):
- We add Figure 7 to appendix D. This figure shows the mean and IQM of the area under the curve of the learning curves for each method, aggregated over seeds and environments, to highlight the sample efficiency of our method.
- We add Figure 10 to appendix F. This figure shows a hyperparameter sensitivity analysis.
- We add Figure 11 to appendix F. This figure shows the result of partial hyperparameter searches, where we try to maximize performance with only 1 or 2 of our hyperparameters
- We add a reference to the IBC paper in our related work.

---

### Meta-Review · Area_Chair_yyXN · 2023-12-06

**Metareview:**

All reviewers found the paper to be tackling an interesting topic, and the paper to be well-written, and useful. No major issues emerged in the discussion.

**Justification For Why Not Higher Score:**

This one could be bumped up if there's space. I don't see any particular reason not to do so. Everyone basically thought the paper was interesting, and there don't seem to be any serious issues with it.

**Justification For Why Not Lower Score:**

All but one reviewer gave scores 6 or above. The only reviewer below 6 gave a 5, and was mostly concerned with sensitivity analysis of hyperparameters, which I don't think is a serious enough concern to cause us to reject the paper. The authors did additional sensitivity analysis in the rebuttal phase anyway.

---

### Decision · Program_Chairs · 2024-01-16

Accept (poster)